# First Records of Possibly Human Pathogenic *Rickettsia* Species in Bat Ticks, *Carios vespertilionis*, in Sweden

**DOI:** 10.3390/microorganisms11020357

**Published:** 2023-01-31

**Authors:** Eszter Tompa, Thomas G. T. Jaenson, Peter Wilhelmsson

**Affiliations:** 1Department of Biomedical and Clinical Sciences, Division of Inflammation and Infection, Linköping University, 581 83 Linköping, Sweden; 2Evolutionary Biology Centre, Department of Organismal Biology, Uppsala University, 752 36 Uppsala, Sweden; 3Department of Clinical Microbiology, Region Jönköping County, 553 05 Jönköping, Sweden

**Keywords:** *Rickettsia* spp., *Carios vespertilionis*, *Pipistrellus pygmaeus*, Sweden

## Abstract

The Soprano pipistrelle bat, *Pipistrellus pygmaeus*, is a common species in large parts of Sweden. Many of its natural habitats are near human habitations. This creates opportunities for ticks infesting these bats to encounter humans and possibly transmit zoonotic pathogens by tick bites. The bats are often infested with *Carios vespertilionis*, a tick species that, in addition to bats, has been recorded to bite humans on occasion. This study aimed to investigate if *C. vespertilionis* acts as a reservoir for *Anaplasma phagocytophilum*, *Neoehrlichia mikurensis*, Tick-borne encephalitis virus, and species of *Babesia* and *Rickettsia* and to improve currently used conventional PCR protocols for molecular species determination of *Rickettsia* spp. Ninety-two *C. vespertilionis* ticks were collected from underneath a bat-box harbouring *P. pygmaeus*. Pathogen-specific PCR assays showed that 58.4% were positive for *Rickettsia* spp. and negative for the other pathogens analysed. Phylogenetic analyses indicate that the species belong to *R. parkeri*, *R. conorii*, *R. slovaca*, *R. sibirica* subsp. *mongolotimonae*, *R. rickettsii*, and a hitherto uncultured *Rickettsia* sp. Several of these species are considered pathogenic to humans. Given the ecology and behaviour of *C. vespertilionis*, it may be a vector of these rickettsiae among bats and occasionally humans. To determine the *Rickettsia* species with certainty, and to determine if *C. vespertilionis* may be a reservoir and vector of the *Rickettsia* spp., further studies are needed.

## 1. Introduction

Due to several recent outbreaks of serious infections caused by bat-associated pathogens, research on such pathogens has increased drastically during the past two decades [1]. Among approximately 1400 described bat species, 19 can be found in Sweden. The Soprano pipistrelle, *Pipistrellus pygmaeus* (Chiroptera; Vespertilionidae), is a common species in southern and south–central Sweden. This species prefers to rest and breed in hollow trees, attics, roof coverings, and wall cavities [2,3]. Many of the resting places of this bat species are near human living spaces, which creates an opportunity for bat ectoparasites, such as ticks, to blood-feed on humans and thus a potential to transmit pathogens to them [4,5].

European populations of *P. pygmaeus* are often infested by ticks, such as the Short-legged bat tick *Carios vespertilionis* (Ixodida; Argasidae). To this date, this is the only argasid species and the only bat-associated species recorded from Sweden [4]. This tick species is usually host-specific to bats but has been described as ‘highly aggressive’, with a few reports of occasional infestations on humans, dogs, and birds [4,5,6].

Several bacterial, viral, and protozoan pathogens have been identified in *C. vespertilionis* ticks. However, the role of this species as a vector and/or a reservoir is still under debate. As far as we know, there is only one study where the prevalence and species of pathogens in *C. vespertilionis* in Sweden have been investigated. The ticks investigated in this study have previously been analysed for *Borrelia* spp. where a high prevalence of *Borrelia* of the relapsing fever group was found [7]. The identified species, *Borrelia* sp. CPB1, has also been identified in *C. vespertilionis* in France and could perhaps pose a threat to *Pipistrellus* bats [7,8]. There have been findings of microorganisms with known and unknown pathogenicity worldwide, as seen in Table 1. However, the prevalence of *Anaplasma phagocytophilum*, *Babesia* spp., Tick-borne encephalitis virus (TBEV), *Neoehrlichia mikurensis*, and *Rickettsia* spp. in *C. vespertilionis* ticks from Sweden has, to our knowledge, not been studied before.

As mentioned, the prevalence of *Borrelia* spp. in *C. vespertilionis* ticks has been reported earlier [7]. The present study investigated the role of *C. vespertilionis* as a potential reservoir of *A. phagocytophilum*, *N. mikurensis*, TBEV, and species of *Babesia* and *Rickettsia*. In particular, we investigated if the molecular species determination of *Rickettsia* spp. in *C. vespertilionis* ticks could be improved by complementing or exchanging the currently used conventional PCR protocols.

## 2. Materials and Methods

### 2.1. Sampling, Analyses, and Processing of Ticks

Full details of the sampling site, procedures for tick collection and identification, extraction of total nucleic acid, and synthesis of cDNA are available in the previous report [7].

In summary, ticks were collected from June to August in 2015 and 2018 from a tray placed underneath a bat box located at Snesslinge, province of Uppland, south–central Sweden (60°19.567 N, 18°15.067 E). In addition, one tick collected from inside a private house at Älmhult, province of Småland, southern Sweden (56°32.720 N, 13°52.667 E) in 2019 was included. All ticks were photographed and morphologically identified to life stage (larva, nymph, adult). Each tick was homogenised individually using a TissueLyser II (Qiagen, Hilden, Germany). The total nucleic acid (NA) was extracted, purified, and isolated using MagAttract^®^ Viral RNA M48 kit in a BioRobot M48 workstation (Qiagen). The eluted NA was reverse-transcribed to cDNA using illustra™ Ready-to-Go RT-PCR Beads kits (GE Healthcare, Amersham Place, UK). The final product was frozen at −80 °C.

### 2.2. Detection of Anaplasma phagocytophilum

A TaqMan real-time PCR assay was used to detect *A. phagocytophilum*. The primers and probe used for the assay amplify a 64 bp long amplicon of the citrate synthase gene (*gltA*) of *A. phagocytophilum* (Table 2). A synthetic plasmid containing the target sequence (nucleotides 304 to 420 of the *A. phagocytophilum gltA* gene (GenBank: AF304137)) was synthesized and cloned into pUC57 vector (GenScript, Piscataway, NJ, USA) and used as a positive control. Further details of the method are described in an article by Henningsson et al. [13].

### 2.3. Detection of Babesia Species

An SYBR green real-time PCR assay was used to detect *Babesia* spp. The primers used for the assay amplify a 411–452 bp long amplicon of the *18S* rRNA gene of *Babesia* (Table 2). A synthetic plasmid containing the target sequence (nucleotides 467 to 955) of the *B. divergens 18S* rRNA gene (GenBank: AJ439713) was synthesized and cloned into pUC57 vector (GenScript) and used as a positive control. Further details on the method are described in an article by Casati et al. [14].

### 2.4. Detection of Neoehrlichia mikurensis

An SYBR green real-time PCR assay was used to detect *N. mikurensis*. The primers used for the assay amplify a 107 bp long amplicon of the *16S* rRNA gene of *N. mikurensis* [15] (Table 2). As a positive control, cDNA samples positive for *N. mikurensis* confirmed by sequencing in an earlier study [15] were used. Further details of the method are described in a previous article by Labbé Sandelin et al. [15].

### 2.5. Detection of TBEV

A duplex TaqMan real-time PCR assay was used for the detection of TBEV, as previously described by Schwaiger and Cassinotti, and Gäumann et al. [16,17]. The primers and probes are designed to target all three subtypes of TBEV to amplify a 68 bp and an 88 bp long amplicon, respectively [16,17] (Table 2).

### 2.6. Detection of Rickettsia and Determination of Species

A TaqMan real-time PCR assay with primers designed to target the *gltA* gene of *Rickettsia* spp. was used to detect *Rickettsia* spp. as previously described [7,18]. The primers and probe used in the assay amplify a 74 bp long amplicon (Table 2) [18]. A synthetic plasmid containing the target sequence (nucleotides 1102 to 1231) of the *R. rickettsii gltA* gene (GeneBank: U59729) was synthesized and cloned into pUC57 vector (GenScript) and used as a positive control.

The samples positive in the TaqMan real-time PCR assay were further analysed using conventional PCR assays where fragments of the genes coding for outer membrane proteins A and B (*ompA*, *ompB*), 17 kDa, *gltA* (I and II), and *16S* rRNA (*rrs)* genes were amplified as previously described (Table 2) [18,19,20,21,22]. The *gltA* gene was amplified both using a conventional PCR assay and a semi-nested PCR assay (Table 2) [21,22]. The PCR products amplified by the conventional PCR assays were sequenced by Macrogen Inc. (Amsterdam, The Netherlands). All sequences were confirmed by sequencing both strands. The obtained chromatograms were edited and analysed using BioEdit v7.0 (Tom Hall, Ibis Therapeutics, Carlsbad, CA, USA). The edited sequences were compared to existing sequences in the Basic Local Alignment Search Tool (BLAST).

### 2.7. Phylogenetic Analyses

Phylogenetic trees were constructed with MEGA11 by maximum likelihood using the Kimura 2-parameter. Complete deletion with a bootstrap value of 500 replicates was used.

## 3. Results

### 3.1. Number of Ticks Collected

A total of 92 ticks were collected in the province of Uppland, south–central Sweden, in the summers of 2015 and 2018. In addition, one specimen was collected in the province of Småland, southern Sweden, in 2019. Of these, 28 ticks were collected in 2015, and 63 were collected in 2018. All ticks were microscopically identified as *C. vespertilionis*. Of all 92 ticks, 48 were identified as nymphs, 31 as larvae, and 13 as adults (Table 3). These results have been reported earlier [7].

### 3.2. Prevalence of Rickettsia

Of the 92 examined ticks, 54 (58.7%) were determined to be *Rickettsia*-positive using the real-time PCR assay, including the adult tick from Småland. Of these 54 ticks, 77% (10/13) were adults, 56.3% (27/48) were nymphs, and 56.7% (17/31) were larvae, as seen in Table 3. Of the ticks collected in 2015, 50% (14/28) were positive for *Rickettsia*, and 62% (39/63) of ticks collected in 2018 were positive for *Rickettsia*.

### 3.3. Sequencing of PCR Products and Phylogenetic Sequence Analysis

The results of the amplification and sequencing of the *ompA*, *ompB*, *rrs*, *gltA* (I), *gltA* (II), and 17 kDa amplicons can be seen in Table 4.

Seven samples could not be analysed using the *ompA*, *rrs*, and the *gltA* (II) PCR protocols due to insufficient volumes of NA. As seen in Table 4, the amplification and sequencing of the *ompA* gene amplicon was successful in 9 of 47 samples, including the tick from Småland. All nine samples, referred to as ‘*Rickettsia* sp. AvBat *ompA*’ in Table 4, showed 100% identity (305/305) with *Rickettsia parkeri* (accession no. MK801772.1).

Further, the amplification and sequencing of the *ompB* gene amplicon was successful in 49 of 54 samples, including the tick from Småland. Of these, 47 sequences, referred to as ‘*Rickettsia* sp. AvBat *ompB*’ in Table 4, were identical to each other and showed 100% sequence identity (250/250) with those of *Rickettsia parkeri* (accession no. KY124259), *Rickettsia slovaca* (accession no. AF123723), *Rickettsia conorii* (accession no. AF149110), and *Rickettsia sibirica* sub-species *mongolotimonae* (accession no. AF123715). One sequence referred to as ‘*Rickettsia* sp. AvBat *ompB* nymph 229 Uppland 2018’ in Table 4, showed 99.6% sequence identity (251/252) with the aforementioned sequences, and the other sequence, referred to as ‘*Rickettsia* sp. AvBat *ompB* nymph 191 Uppland 2018’ in Table 4, showed 98.8% sequence identity (249/252) with the aforementioned sequences.

Four of the forty-seven samples could be successfully amplified and sequenced using the *rrs* gene amplicon. Two samples referred to as ‘*Rickettsia* sp. AvBat *rrs*’ in Table 4 were identical to each other and showed 100% sequence identity (1356/1356) to an uncultured *Rickettsia* species (accession no. MG827267.1). One sample referred to as ‘*Rickettsia* sp. AvBat *rrs* nymph 159 Uppland 2018’ showed a 99.4% sequence identity (1248/1256) with the aforementioned sequence. Finally, one sample referred to as ‘*Rickettsia* sp. AvBat *rrs* nymph 161 Uppland 2018’ showed a 94.7% sequence identity (1325/1399) with *Rickettsia conorii* (accession no. NR_074480.2).

One *gltA* (II) gene amplicon could be successfully sequenced. This sample, referred to as ‘*Rickettsia* sp. AvBat *gltA* adult 179 2018 Uppland’ in Table 4, showed 100% sequence identity (333/333) with *Rickettsia parkeri* (accession no. MK814825.1).

Amplification and sequencing of the *gltA* (I) gene amplicon was successful in 43 samples, including the tick from Småland. All sequences referred to as ‘*Rickettsia* sp. AvBat *gltA*’ in Figure 1 and Table 4 were identical to each other and showed 100% sequence identity (795/795) with *Rickettsia parkeri* (accession no. MN388794).

Amplification and sequencing of the 17 kDa gene amplicon was successful in 32 samples, including the tick from Småland. All sequences referred to as ‘*Rickettsia* sp. AvBat 17 kDa’ in Figure 2 and Table 4 were identical to each other and showed a 100% sequence identity (434/434) with *Rickettsia rickettsii* (accession no. DQ176856.1).

All consensus sequences obtained and used for BLAST search in this study can be found in Appendix A.

### 3.4. Prevalence of Other Tick-Borne Microorganisms

All ticks, including the tick from Småland, were negative for *A. phagocytophilum*, *N. mikurensis*, *Babesia* spp., and TBEV.

### 3.5. Co-Infections

A high prevalence of species belonging to the relapsing fever *Borrelia* complex has been described in a previous report [7]. Of the ticks collected from below the bat box, 16.5% (15/91) were co-infected with *Borrelia* sp. and *Rickettsia* sp. We collected 3 of these ticks (2 nymphs and 1 adult) in 2015 and 12 ticks (8 larvae and 4 nymphs) in 2018. The tick from Småland was co-infected with *Borrelia* sp. and *Rickettsia* sp.

## 4. Discussion

The only argasid tick species recorded in Sweden is *C. vespertilionis.* This species is also the only known bat-associated tick species in Sweden, making it a highly interesting species to investigate [4]. There is, as far as we know, no previous Swedish record of any *Rickettsia* species from this tick species or from any bat species in Sweden. *Rickettsia* spp. were identified in almost two-thirds (58.7%) of the investigated ticks. The phylogenetic analyses of the genes *gltA* and 17 kDa, and the sequencing of the amplicons from *ompA*, *ompB*, and *rrs* indicate that the specific species identified in this study are phylogenetically close to *R. parkeri*, *R. conorii*, *R. slovaca*, *R. sibirica* subsp. *mongolotimonae*, *R. rickettsii*, and an uncultured *Rickettsia* sp. The pre-dominant *Rickettsia* species found in *I. ricinus* ticks in Sweden is *R. helvetica* [21,23]. Moreover, there is a single record of *R. sibirica* from *I. ricinus* [21]. In addition, *R. felis* infections have been diagnosed in two human patients with sub-acute meningitis [24]. *R. aeschlimannii* has been identified in adult *Hyalomma* ticks in Sweden. However, members of this tick genus are not (yet) considered to be permanently present in Sweden [23].

The species *R. conorii*, *R. slovaca*, and *R. sibirica* subsp. *mongolotimonae* are under surveillance in many European countries as they are pathogenic to humans [25]. Sub-species of *R. conorii* cause Mediterranean spotted fever (MSF). In European patients, fatality or complications have been seen in up to 32% of cases. They are caused by different sub-species of R. conorii and involve many medical factors such as delayed and incorrect treatment, delayed diagnosis, and co-morbidities [25,26,27]. *R. slovaca* is one of the causative species of Scalp Eschar and Neck Lymphadenopathy After Tick Bite (SENLAT), a disease associated with several painful symptoms and fatigue which can become chronic [26,28]. To this date, there are at least 27 reports of human cases of *R. sibirica* subsp. *mongolotimonae* rickettsioses [26]. Infections are usually mild, but cases with acute renal failure, retinal vasculitis, hyponatremia, and lethargy have been reported [29]. *R. rickettsii* and *R. parkeri* are currently not under surveillance in any European country [25]. Therefore, information on any European cases of rickettsioses caused by these species is scarce. There have been around 15 confirmed human cases of *R. parkeri* rikettsiosis in the Western hemisphere between 2004 and 2013, all with mild clinical outcomes [28]. For this reason, *R. parkeri* might not require extensive surveillance. *R. rickettsii*, however, is the causative of Rocky Mountain Spotted Fever (RMSF) in the Americas. For reasons yet to be fully understood, the fatality rates of RMSF varies widely among different strains in the US, and there have been outbreaks in Mexico and Panama with rates as high as 38% and 100% [28].

The known vectors for *R. conorii* are *Dermacentor* spp. and *Rhipicephalus* spp. [6,25,26,27]. The vectors for *R. sibirica* subsp. *mongolotimonae* are *Hyalomma* spp. and *Rhipicephalus pusillus*. However, the vectors in Europe are unknown [6,25,26]. There are studies that implicate the involvement of *R. sanguineus* s.l. in the transmission of *R. conorii* and *R. rickettsii* [6]. Species of *Dermacentor* and *Amblyomma* and are known vectors for the *R. rickettsii* in the Americas [28]. The vectors of *R. slovaca* are *D. marginatus* and *D. reticulatus* [6,25,26,27]. Lastly, the vectors of *R. parkeri* are *D. variabilis* and species of *Amblyomma* [28].

As seen in Table 1, there have been findings of *Rickettsia* spp. in *C. vespertilionis* ticks. Notably, *R. rickettsii* was identified in *C. vespertilionis* ticks in China [12]. This information and the findings of our study indicate that this tick species may have a role as a vector and/or reservoir for *Rickettsia* sp. in Sweden. Thorough further investigations on more tick specimens from different geographic areas and possibly from different mammalian hosts are needed to confirm this.

Rickettsioses are among the oldest known vector-borne diseases. Knowledge about their medical importance has increased dramatically during the past 25 years [26,30]. There are currently 29 acknowledged species of the *Rickettsia* genus. The species are phylogenetically divided into separate groups: the spotted fever group (SFG), the typhus group (TG), the *R. bellii* group, and the *R. canadensis* group [30]. The TG consists of the species *R. typhi* and *R. prowazekii*, with fleas and body lice as their respective vectors [25]. Epidemic typhus, caused by *R. prowazekii*, is not considered to be present in Europe [25,26]. Murine typhus, caused by *R. typhi*, is not present in Sweden, but a few imported cases occur each year [31]. The SFG contains the majority of all species that cause disease in humans. The more than 20 species are primarily transmitted by many different tick species, such as *Rhipicephalus* spp., *Ixodes* spp., and *Dermacentor* spp. [25,26,28,30].

As the knowledge on rickettsiae has increased, many species that were initially classified to the order Rickettsiales have been moved to other orders [26,28]. The guidelines for classification and species determination of the genus *Rickettsia* are an ongoing discussion among rickettsiologists around the world [26,28]. Cultivation of rickettsiae is limited to tissue cultures, guinea pigs, and embryonated chicken eggs as these bacteria are strictly intra-cellular. Staining is also more challenging as regularly used methods cannot be used [30]. More recently proposed guidelines for taxonomic classification of *Rickettsia* are sequencing of the genes *rrs*, *gltA*, *ompA*, *ompB*, and *gene D* and comparison of these to validated *Rickettsia* species [28,30,32]. The *rrs* gene has been suggested as the best tool for classification on a genus level and the *ompA* gene for classification into the SFG. In the case of the absence of *ompA*, these samples may still classify as a SFG *Rickettsia* if they show a certain degree of similarity with two of the genes *rrs*, *gltA*, *ompB*, or gene D from a validated *Rickettsia* species [32].

Based on this, we added three other conventional PCR protocols for *rrs*, *gltA*, and *ompA*, which had been created for and tested on *Ixodes pacificus* ticks [22], to our existing protocols. It was deemed that SFG *Rickettsia* was the most likely group of *Rickettsia* to be found in *C. vespertilionis* ticks. The TG species are transmitted by lice and fleas and, as mentioned above, are not present in Sweden [30,31]. While *R. bellii* is transmitted by many tick species, it has only been recorded in the Americas [28].

In terms of the number of successfully sequenced samples, the added protocols performed poorly compared with *gltA* (I), *ompB*, and 17 kDa. The analyses for *gltA* (I), *ompB*, and 17 kDa were performed in 2020, whereas the three additional genes were analysed in 2022. All used NAs were extracted at the same time and were stored in a −80 °C freezer. The cDNA was synthesised at several times over the years, last in 2022 when the analyses for *rrs*, *gltA* (II), and *ompA* were performed. Long-term storage of RNA and repeated freeze–thawing cycles can cause significant degradation [33,34]. We believe this may be the cause of the poor performance and suggest that, in the future, all cDNA should be synthesised shortly after the extraction of NA as it has increased stability compared to RNA. There is also the possibility of co-infections with several species of *Rickettsia* that cause interference in the obtained chromatograms from the sequencing. This cannot be investigated any further with the methods used in this study and is therefore of interest for future studies.

Several putative tick-borne pathogens, such as *Borrelia* sp., *Ehrlichia* sp., *Babesia* sp., and *Bartonella* sp., have been molecularly identified in *C. vespertilionis* ticks in Europe in previous studies [7,8,9,10,11]. However, with the use of molecular assays, it is not possible to distinguish if the genomes of the pathogens identified were viable nor if the genomes came from the tick or the bat host. As previously reported, nearly all investigated ticks contained host blood [7]. Therefore, the results of this study could indicate what microorganisms these bats are carrying. Obtaining ethical approvals for sampling *P. pipistrellus* bats in Sweden may be difficult as all bat species in Sweden are protected by law [35]. As earlier stated, *P. pygmaeus* bats often live in close proximity to humans. This creates an opportunity for *C. vespertilionis* ticks to bite humans and possibly transmit pathogens. For these reasons, it is of particular interest to investigate what pathogens *C. vespertilionis* contain.

In this study, we found indications of several human pathogenic species of *Rickettsia* in the investigated ticks. Our results reflect that the taxonomic classification on a species level of *Rickettsia* is often faced with difficulties. Due to the fact that several of the species pose a potential danger to humans, it is of importance to further investigate the host associations of the *Rickettsia* species, their potential to be transmitted horizontally and/or vertically by *C. vespertilionis*, and their geographical distribution.

## 5. Conclusions

The ticks investigated contain a high prevalence of *Rickettsia* species. Several of the species possibly belong to the SFG, which contains human-pathogenic species. Further studies are needed to determine with certainty the species identity of these rickettsiae species detected in *C. vespertilionis* and to determine if this tick species is a reservoir and vector of these rickettsiae.

## Figures and Tables

**Figure 1 microorganisms-11-00357-f001:**
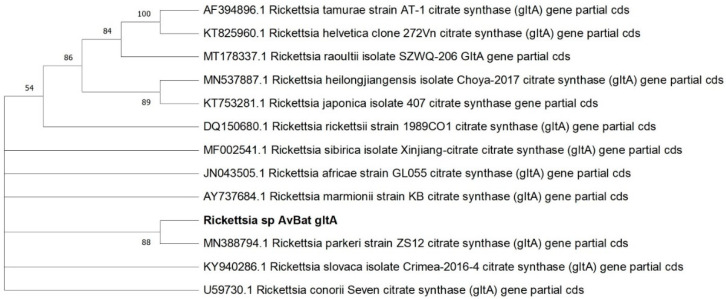
Phylogenetic tree based on *gltA* (I) gene sequences of *Rickettsia* species, constructed by maximum likelihood using Kimura 2-parameter and complete deletion. Sequences detected in our study (‘*Rickettsia* sp. AvBat *gltA*’, n = 43) are highlighted in bold. The reliability of the tree was tested by 500 bootstrap replicate analyses; only values greater than 50% are shown.

**Figure 2 microorganisms-11-00357-f002:**
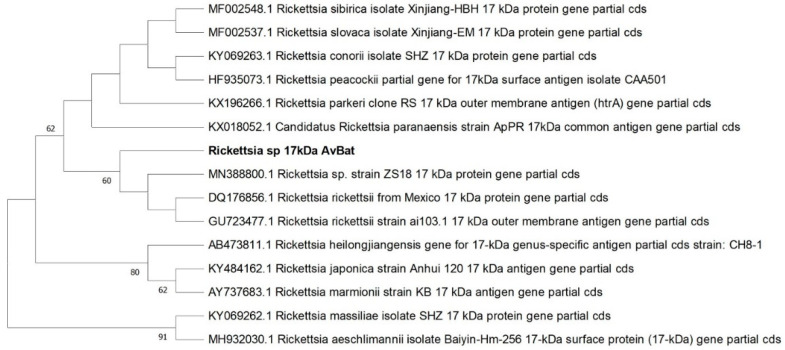
Phylogenetic tree based on 17 kDa gene sequences of *Rickettsia* species, constructed by maximum likelihood using Kimura 2-parameter and complete deletion. Sequences detected in our study (‘*Rickettsia* sp. AvBat 17 kDa’ n = 32) are highlighted in bold. The reliability of the tree was tested by 500 bootstrap replicate analyses; only values greater than 50% are shown.

**Table 1 microorganisms-11-00357-t001:** Bacteria identified in *C. vespertilionis* ticks collected from different countries.

Pathogen	Species	Country	Reference(s)
*Borrelia*	*Borrelia* sp. CPB1	France, Sweden	[7,8]
*Bartonella*		Hungary	[9]
*Babesia*		Hungary, Romania, Italy, Kenya, Vietnam, China Finland, UK	[10] [11]
*Ehrlichia*	*Ehrlichia* sp. AvBat *Ehrlichia* spp.	France United Kingdom	[8] [11]
*Rickettsia*	*Rickettsia* sp. AvBat	France	[8]
	SFG *Rickettsia*	United Kingdom	[11]
	*Rickettsia helvetica*	China	[9]
	*Rickettsia africae*-like	Hungary	[9]
	*Rickettsia raoultii*, *Rickettsia rickettsii*	China	[12]

**Table 2 microorganisms-11-00357-t002:** Primers and probes used for molecular analysis of tick-borne pathogens.

Organism	Target	Primer/Probe Name	Sequence (5′-3′)	Amplicon Length (bp)	References
*A. phagocytophilum*	*gltA*	ApF	TTTTGGGCGCTGAATACGAT	64	[13]
		ApR	TCTCGAGGGAATGATCTAATAACGT		
		ApM	FAM-TGCCTGAACAAGTTATG-BHQ1		
*Babesia* spp.	*18S* rRNA	BJ1	GTCTTGTAATTGGAATGATGG	411–452	[14]
		BN2	TAGTTTATGGTTAGGACTACG		
*N. mikurensis*	*16S* rRNA	Neo_16S_F	GTAAAGGGCATGTAGGCGGTTTAA	107	[15]
		Neo_16S_R	TCCACTATCCTCTCTCGATCTCTAGTT TAA		
	11,054–1112 ^a^	F-TBE 1	GGGCGGTTCTTGTTCTCC	68	[16]
TBEV		R-TBE 1	ACACATCACCTCCTTGTCAGACT		
		TBE-probe-WT	FAM-TGAGCCACCATCACCCAG ACACA-BHQ1		
	1329–1416 ^a^	TBEE-F6	GGCTTGTGAGGCAAAAAAGAA	88	[17]
		TBEE-R2	TCCCGTGTGTGGTTCGACTT		
		TBEE-P4	HEX-AAGCCACAGGACATGTGTACG ACGCC-BHQ1		
*Rickettsia* spp.	*gltA*	CS-F	TCGCAAATGTTCACGGTACTTT	74	[18]
		CS-R	TCGTGCATTTCTTTCCATTGTG		
		CS-P	FAM-TGCAATAGCAAGAACCGTAGG CTGGATG-BHQ1		
	17 kDa	Rr17kDa.61p	GCTCTTGCAACTTCTATGTT	434	[19]
		Rr17kDa.492n	CATTGTTCGTCAGGTTGGCG		
	*ompB*	Rc.rompB.4362p	GTCAGCGTTACTTCTTCGATGC	475	[20]
		Rc.rompB.4836n	CCGTACTCCATCTTAGCATCAG		
		Rc.rompB.4496p	CCAATGGCAGGACTTAGCTACT	267	
		Rc.rompB.4762n	AGGCTGGCTGATACACGGAGTAA		
	*gltA (I)*	RH314	AAACAGGTTGCTCATCATTC	832	[21]
		CSF-R	AAGTACCGTGAACATTTGCGA		
		CS-Ric-R	CAGTGAACATTTGCGACGGTA		
		CS535d	GCAATGTCTTATAAATATTC	SP	
	*gltA (II)*	Forward	GGCTAATGAAGCGGTAATAA ATATGCTT	341	[22]
		Reverse	TTTGCGACGGTATACCCATAGC		
	*ompA*	Forward	CACYACCTCAACCGCAGC	438–444	[22]
		Reverse	AAAGTTA TATTTCCTAAACCYGTATAAKTATCRGC		
	*16S* rRNA	Forward	TAAGGAGGTAATCCAGCC	1482–1483	[22]
		Reverse	CCTG GCTCAGAACGAA		

^a^ = Genome region of TBEV strain Neudoerfl (U27495) that is present in all three subtypes of TBEV. Abbreviations: FAM, 6-carboxyfluorescein; HEX, 5(6)-carboxyfluorescein; BHQ, Black Hole Quencher; SP, sequencing primer.

**Table 3 microorganisms-11-00357-t003:** Tick-borne microorganisms detected in *Carios vespertilionis* ticks from the province of Uppland, in 2015 and 2018, and from the province of Småland, in 2019.

Developmental Stage	No. of Examined Ticks	*Rickettsia* spp.	*A. phagocytophilum*	*N. mikurensis*	*Babesia* spp.	TBEV
Larva	31	17 (56.7)				
Nymph	48	27 (56.3)				
Adult ^a^	13	10 (77.0)				
Total	92	54 (58.7)	0	0	0	0

^a^ = One adult *C. vespertilionis* tick in this group was collected in the province of Småland.

**Table 4 microorganisms-11-00357-t004:** Number of specimens with successful amplification of *Rickettsia* gene targets by conventional PCR assays. The sequences obtained from the amplification of the 17 kDa and *gltA* (I) genes have been used for phylogenetic assays.

Gene	No. of Successfully Sequenced Samples	Sample	Species	Identity
17 kDa	32/54	*Rickettsia* sp. AvBat 17 kDa	*R. rickettsii*	100% (434/434)
*gltA (I)*	43/54	*Rickettsia* sp. AvBat *gltA*	*R. parkeri*	100% (795/795)
*ompA*	9/47 *	*Rickettsia* sp. AvBat *ompA*	*R. parkeri*	100% (305/305)
*ompB*	49/54	*Rickettsia* sp. AvBat *ompB*	*R. parkeri*, *R. slovaca*, *R. conorii*, *R. sibirica* subsp. *mongolotimonae*	100% (250/250)
		*Rickettsia* sp. AvBat *ompB* nymph 229 Uppland 2018	*R. parkeri*, *R. slovaca*, *R. conorii, R. sibirica* subsp. *mongolotimonae*	99.6% (251/252)
		*Rickettsia* sp. AvBat *ompB* nymph 191 Uppland 2018	*R. parkeri*, *R. slovaca*, *R. conorii, R. sibirica* subsp. *mongolotimonae*	98.8% (249/252)
*gltA (II)*	1/47 *	*Rickettsia* sp. AvBat *gltA* adult 179 Uppland 2018	*R. parkeri*	100% (333/333)
*rrs*	4/47 *	*Rickettsia* sp. AvBat *rrs*	uncultured *Rickettsia* species	100% (1256/1356)
		*Rickettsia* sp. AvBat *rrs* nymph 159 Uppland 2018	uncultured *Rickettsia* species	99.4% (1248/1256)
		*Rickettsia* sp. AvBat *rrs* nymph 161 Uppland 2018	*R. conorii*	94.7% (1325/1399)

* Seven samples could not be analysed due to insufficient volumes of NA.

## Data Availability

The data supporting the conclusions of this article are included within the article. Raw data can be shared with researchers upon request.

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
