# Peer review of "First Records of Possibly Human Pathogenic Rickettsia Species in Bat Ticks, Carios vespertilionis, in Sweden"

_microorganisms, 2023, doi:10.3390/microorganisms11020357_

Round 1
Reviewer 1 Report
The mansucript “First records of human pathogenic Rickettsia species in bat ticks, Carios vespertilionis in Sweden” is well writing, and in general, the methods used are correct and the methodology is clearly described, however, I have reservations regarding phylogenetic analyzes.
In my opinion, the trees are not informative, they don't contribute anything that the BLAST analysis shows. (1) Trees with low points of support for branches do not say much, it proves that such a position on the tree is accidental don’t show the true relationships. (2) Analysis provided based on ompB gen cannot show clear results/relationships because the used fragment is too short (about 250 bp), it is good for detection but not for taxonomic or phylogenetic identification (which is shown by the results presented in the publication). (3) Trees based on rrs gene have so many polytomies, is not look good.
BLAST analysis is well done so I suggest leaving one maybe two trees, the ones that best reflect the results of the comparative analysis of sequences, and abandoning the others. And I suggest trying to perform phylogenetic analyses using ML for all of them ( Did Authors check the best model for each analysis independently?), boosting the bootstraps, and well-rooted trees.
Minor remark: I suggest rewrite a bit the sentence in line 139 “and on one occasion in the province…”.
Author Response
Based on this reviewer's comments, we have removed all but two phylogenetic trees based on the length of the fragments and the number of successful amplifications. The ones kept are from gltA (I) (795 bp) where 43 of 54 were successfully amplified, and 17kDa (434 bp) where 32 of 54 samples were successfully amplified. The analysis of these fragments has been done using maximum-likelihood instead.
The sentence in question has been rewritten. See page 5 and lines 136-137 in the reviewed manuscript.
Reviewer 2 Report
The study by Tompa et al. investigates the occurrence of tick-borne pathogens in the bat soft tick Carios vespertilionis. The methodology is up-to-date and the results are important and interesting. The manuscript is well-written but need some editing before its publication can be suggested, as outlined below.
- lines 30-31: please remove word repetition from the first sentence (the word pathogen/pathogenic is repeated three times)
- lines 40-41: the name of C. vespertilionis is not “Short-legged bat tick” in English, please remove
- line 44: tick-infestation should not be called as “attack” in scientific texts
- lines 52-53: Pipistrellus as a genus name should be in italics
- line 59: please delete “was”
- Table 1: in the row “Babesia” the study by Lv et al. (No. 11) is referred to but the UK is not mentioned among the countries
- in all phylogenetic trees the species and genus names should be in italics, but to request this is a decision by the journal editors
- lines 179, 189 and elsewhere: the genus name in Rickettsia sp. AvBat should be in italics
- line 301: “rickettsiae” is a collective term (not a taxonomic category), therefore should start with lower case letter
Author Response
- Lines 30-31: please remove word repetition from the first sentence (the word pathogen/pathogenic is repeated three times)
The sentences have been rephrased to avoid repetition. See page 1, lines 30-31 in the revised manuscript. - Lines 40-41: the name of C. vespertilionis is not “Short-legged bat tick” in English, please remove
The English name of vespertilionis is, in fact, Short-legged bat tick. - Line 44: tick-infestation should not be called as “attack” in scientific texts
Changed to “infestations”. See page 1, line 43. - Lines 52-53: Pipistrellus as a genus name should be in italics.
Changed to be in italics. See page 2, line 51. - Line 59: please delete “was”
Word has been removed. See page 2, line 57. - Table 1: in the row “Babesia” the study by Lv et al. (No. 11) is referred to but the UK is not mentioned among the countries
The UK has been added to the table. See page 2. - In all phylogenetic trees the species and genus names should be in italics, but to request this is a decision by the journal editors
The software used does not allow this form of editing, unfortunately. - Lines 179, 189 and elsewhere: the genus name in Rickettsia sp. AvBat should be in italics.
Has been changed throughout the revised manuscript. - Line 301: “rickettsiae” is a collective term (not a taxonomic category), therefore should start with lower case letter.
Has been changed. See page 9, line 280.
Reviewer 3 Report
The article provides new information about Rickettsia in argasid ticks in Sweden. However, the authors' conclusions are not supported by the results. Perhaps, different phylogenetic analyses could provide better results.
Title:
Based on the study it is not possible to stand that the Rickettsia species found in C. vespertilionis are pathogenic.
Materials and Methods:
Phylogenetic analyses:
Concatenated fragments of the evaluated genes should be used to obtain a more accurate diagnosis of the Rickettssia species.
Results:
It is important to include the accession numbers of all sequences
It is not clear to which ticks samples the sequences obtained for each of the genes refer, since, for some genes, more than one sequence was obtained (ompB and rrs). Authors should consider including a table containing information about Rickettsia-positive samples.
Discussion:
L247-250: Based on phylogenetic analyses performed in this study it is not possible to make this statement.
Conclusions:
L350: Not every Rickettsia species belonging to the SFG is known to be pathogenic.
Based on the results, it is only possible to say that Rickettsia sp. DNA was found in the evaluated ticks.
Author Response
- Title: Based on the study it is not possible to stand that the Rickettsia species found in vespertilionis are pathogenic.
The title has been changed to “First records of possibly human pathogenic Rickettsia species in bat ticks, Carios vespertilionis in Sweden” in order to better match the findings and conclusions of this study. - Materials and Methods
Phylogenetic analyses: Concatenated fragments of the evaluated genes should be used to obtain a more accurate diagnosis of the Rickettssia
Based on this comment and comments from another reviewer, we have decided to only keep two phylogenetic trees. These trees are based on the gltA gene and 17kDa gene, respectively. - Results: It is important to include the accession numbers of all sequences.
Our sequences obtained in this study can be found in the attached appendix (Appendix 1). - It is not clear to which ticks samples the sequences obtained for each of the genes refer, since, for some genes, more than one sequence was obtained (ompB and rrs). Authors should consider including a table containing information about Rickettsia-positive samples.
The information on the Rickettsia-positive samples is presented in table 4. In the text, it is stated that sequences that are 100% identical to each other are presented under a collective name instead of presenting each sequence individually. - Discussion: L247-250: Based on phylogenetic analyses performed in this study it is not possible to make this statement.
As we state in the sentence, the results indicate that these could be the species mentioned. We do not state that this is a proven fact, therefore we have chosen to keep the sentence as it is. - Conclusions: L350: Not every Rickettsia species belonging to the SFG is known to be pathogenic.
The sentence has been rephrased to clarify that only some species of the SFG are pathogenic to humans. See page 10, lines 331-332 in the revised manuscript. - Based on the results, it is only possible to say that Rickettsia DNA was found in the evaluated ticks.
It is stated in the conclusions that the only thing that can be stated with certainty is that the ticks contain Rickettsia sp. The species are possibly SFG, but as we state further studies are needed to determine the species with certainty. The changed title should also aid in the interpretation of the results.
Round 2
Reviewer 3 Report
Authors addressed most of suggestions made in the original version and improved tables and figures. Below you will find some suggestions and minor comments.
Table 4: Authors should consider presenting in the “identity” column the absolute numbers of aligned residues within parentheses, e.g. 97,78% (397/406).
L 293-296: It would be more appropriate to say that the Rickettsia species found in this study are phylogenetically close to the mentioned species:
“The phylogenetic analyses of the genes gltA, and 17kDa, and the sequencing of the amplicons from ompA, ompB, and rrs indicate that the specific species identified in this study are phylogenetically close to R. parkeri, R. conorii, R. slovaca, R. sibirica subsp. mongolotimonae, R. rickettsii, and an uncultured Rickettsia sp.”
L329-330: Species of Demacentor and Amblyomma are known to be vectors of R. rickettsii in the Americas.
L 355: Amblyomma spp. ticks are also important vectors of SFG Rickettsia.
Author Response
Table 4: Authors should consider presenting in the “identity” column the absolute numbers of aligned residues within parentheses, e.g. 97,78% (397/406).It has been added to table 4 in the revised manuscript.
L 293-296: It would be more appropriate to say that the Rickettsia species found in this study are phylogenetically close to the mentioned species: “The phylogenetic analyses of the genes gltA, and 17kDa, and the sequencing of the amplicons from ompA, ompB, and rrs indicate that the specific species identified in this study are phylogenetically close to R. parkeri, R. conorii, R. slovaca, R. sibirica subsp. mongolotimonae, R. rickettsii, and an uncultured Rickettsia sp.”It has been rephrased accordingly, see Lines 228-229 in the revised manuscript.
L329-330: Species of Demacentor and Amblyomma are known to be vectors of R. rickettsii in the Americas.The sentence has been edited, see line 261 in the revised manuscript.
L 355: Amblyomma spp. ticks are also important vectors of SFG Rickettsia.
Amblyomma spp. are already mentioned as vectors for the SFG Rickettsia R. rickettsii and R. parkeri in lines 260-262.